# Participation and Adherence to Prehabilitation Programs for Colorectal Cancer

**DOI:** 10.3390/nu17111792

**Published:** 2025-05-25

**Authors:** Misha A. T. Sier, Eva Godina, Omar Mollema, Maud P. P. J. Cox, Thais T. T. Tweed, Jan Willem Greve, Jan H. M. B. Stoot

**Affiliations:** 1Department of Surgery, Zuyderland Medical Centre, 6162 BG Sittard, The Netherlands; grevejw@gmail.com (J.W.G.); j.stoot@zuyderland.nl (J.H.M.B.S.); 2School of Nutrition and Translational Research in Metabolism (NUTRIM), Maastricht University, 6229 ER Maastricht, The Netherlands; 3Faculty of Health, Medicine and Life Science, Maastricht University Medical Center+, 6229 ER Maastricht, The Netherlands; godinaevaa@gmail.com (E.G.);; 4Department of Internal Medicine, Máxima Medical Centre, 5504 DB Veldhoven, The Netherlands; m.cox@mmc.nl; 5Department of Surgery, Maastricht University Medical Centre+, 6229 HX Maastricht, The Netherlands; thais_tweed@hotmail.com

**Keywords:** prehabilitation, colorectal cancer, participation, adherence

## Abstract

**Background/Objectives**: The preoperative improvement of patients’ functional capacity (prehabilitation) has gained attention in the surgical field, especially for colorectal cancer (CRC) patients. Despite the recognized benefits of prehabilitation programs, patients’ motivation to participate in and adhere to them remains a significant challenge. Several studies reported difficulties in recruiting participants and low adherence rates. This systematic review explored patients’ motivation for participation and adherence to prehabilitation programs for colorectal cancer surgery. **Methods**: A systematic search was conducted in the PubMed (MEDLINE), Embase, Cochrane, CINAHL, and Web of Science databases. Eligible studies included clinical trials published from inception until December 2024, written in English or Dutch, describing barriers and/or motivators affecting patient participation and adherence in prehabilitation programs. **Results**: A total of 89 studies, including 34 randomized controlled trials, were included. In total, 13,383 patients were included, with 7162 in the prehabilitation cohort. Participation rates ranged from 0 to 99.4%, and adherence rates ranged from 15% to 100%. Factors limiting participation included logistical issues and a busy schedule. Professional guidance, peer support, and regaining a sense of control improved adherence. Medical reasons, conflicting obligations, and intensive exercise limited adherence. **Conclusions**: This systematic review analyzed the current literature on participation and adherence in prehabilitation programs for colorectal cancer surgery patients. Overcoming logistical barriers and patient concerns through flexible, patient-centered approaches may improve participation and adherence. Future research should focus on large-scale randomized controlled trials, diverse healthcare settings, and strategies to enhance engagement with prehabilitation programs.

## 1. Introduction

In recent years, the number of studies on the effect of prehabilitation—the preoperative improvement of patients’ functional capacity—has expanded rapidly [1,2,3,4,5]. However, patients’ motivation to participate in and their adherence to prehabilitation programs remain associated with significant challenges. Multiple studies reported difficulties in recruiting participants, highlighting these barriers [6,7].

Despite major improvements in perioperative care, such as minimally invasive surgery and enhanced recovery protocols, morbidity rates after colorectal surgery remain as high as 40% [8,9,10]. Functional capacity has been identified as a key determinant for a favorable outcomes after surgery [2]. As a result, there is a strong impetus for clinicians to improve patients’ functional capacity [5]. Prehabilitation allows patients to improve their functional capacity before surgery, improving their ability to withstand the upcoming stress of a surgical procedure [11,12]. Recent studies have demonstrated the potential of multimodal prehabilitation to improve patients’ physical status, reduce the length of hospital stay (LOS), and potentially lead to cost savings [1,5,13,14,15]. Early assessment by a nutritionist during the preoperative period is increasingly recommended as part of standard prehabilitation pathways [16].

In the Netherlands, several hospitals have implemented a prehabilitation protocol [17]. However, no uniform nationwide prehabilitation protocol has been established. For prehabilitation to become the standard of care in clinical practice, recruitment and adherence rates need to increase. This remains challenging, particularly since frail, elderly patients, often suffering from decreased physical reserves and adaptive capacity at baseline, represent the primary population of interest [2,18,19,20]. Notably, many colorectal cancer (CRC) patients (55%) are aged 70 years or older [21]. This demographic shift is partly attributable to the widespread implementation of CRC screening programs. These programs allow for the early detection and removal of tumors, which reduces the overall number of advanced cases. As a result, the remaining patient population tends to be older and often presents with more complex medical issues [22]. Furthermore, participation in prehabilitation programs requires discipline, flexibility, and intrinsic motivation, which presents both practical and motivational challenges for patients [23,24,25,26].

Given the significant increase in prehabilitation studies, updating the literature on factors improving participation and adherence in prehabilitation is essential for generating robust data and facilitating its implementation.

### Objective

This study aims to explore how to improve patients’ participation in and adherence to prehabilitation programs for colorectal cancer. To achieve this, the available literature will be systematically reviewed.

## 2. Materials and Methods

This systematic review was prospectively registered in the PROSPERO database (CRD42023481740). The review was conducted following the guidelines described in the Cochrane Handbook for Systematic Reviews of Interventions, Version 6.3, 2022, and the Preferred Reporting Items for Systematic Reviews and Meta-analysis of Individual Participant Data (PRISMA) guidelines [27,28].

### 2.1. Eligibility Criteria

#### 2.1.1. Types of Studies

All clinical trials describing barriers and/or motivators of patient participation or adherence in prehabilitation programs for colorectal cancer surgery were eligible for inclusion. To ensure the inclusion of all relevant articles, studies were not restricted on study type, blinding, or sample size. However, studies were required to be published in peer-reviewed indexed journals.

#### 2.1.2. Participants

Studies involving patients aged ≥ 18 years undergoing elective surgery for colorectal cancer, with or without (neo)adjuvant therapy, were eligible for inclusion. Articles that focus on patients with cancers other than colorectal cancer were excluded.

#### 2.1.3. Interventions

There is no consensus on the design or content of prehabilitation programs and interventions. For this review, we focused on trials with a preoperative prehabilitation program of at least 7 days, involving exercise and/or nutritional interventions. The length of prehabilitation program was chosen based on a recently published Cochrane systematic review [5].

#### 2.1.4. Outcome Measure

The primary outcome parameter in this study was the participation rate. Secondary outcome parameters included motivators and barriers for participation, adherence rates, and patients’ motivators and barriers for adherence. Despite the focus on the participation and adherence of patients in prehabilitation programs, we did not exclude relevant studies not describing both measures.

Baseline characteristics included:Characteristics of the prehabilitation program: interventions (e.g., exercise, nutritional or psychological guidance), length of the program, motivators, and barriersCharacteristics of the patients: baseline characteristics and location of cancerCharacteristics of surgical/oncological treatment: type of surgery, type of (neo)adjuvant treatment, postoperative outcomes such as length of hospital stay (LOS), and overall complications, these will be graded by the Clavien–Dindo classification [29] and further divided into both major (e.g., abdominal sepsis), and minor complications (e.g., wound infection), readmissions within 90 days after surgery.

### 2.2. Search Method for Identification of Studies

#### Literature Search

A systematic literature search in five key healthcare databases was conducted. Pubmed (MEDLINE), Embase, Cochrane, CINAHL, and Web of Science were searched from inception until 31 December 2024. The following search terms were used: prehabilitation, digestive surgery, digestive cancer, gastrointestinal surgery, and gastrointestinal cancer. The search strategy was developed following the guidelines described in the Cochrane Handbook for Systematic Reviews of Interventions, Version 6.3, 2022 [27]. The full search strategy is provided as Appendix A. All references from the four search databases were imported into Rayyan Bibliographic software [30]. A comprehensive cross-reference analysis of identified articles was conducted to ensure all relevant studies and sources were identified and included.

### 2.3. Data Collection and Analysis

#### 2.3.1. Selection of Studies

The titles, abstracts, and descriptor terms of all downloaded material from the electronic searches were read by one researcher (MS), all irrelevant reports were excluded. All citations identified were independently assessed by two researchers (MS and OM) to determine the relevance of each article based on the pre-specified criteria. In cases of uncertainty regarding the relevance of the study, the full article was obtained. Studies were evaluated for relevance based on the study design, types of participants, types of interventions, and outcome measures. After identifying relevant articles, MS and OM independently applied the inclusion criteria. Differences were resolved by discussion with a third reviewer, JS, and a consensus among all reviewers was reached. The selection process is outlined in a flowchart according to PRISMA guidelines [28]. All identified studies were listed in the “Characteristics of Included Studies” table including an evaluation of whether the studies fulfilled the inclusion criteria. Excluded studies and the reasons for exclusion were listed as well.

#### 2.3.2. Data Extraction and Management

All data were extracted independently by three researchers (EG, OM, and MS). The three types of studies we focused on were randomized control trials, observational studies, and qualitative studies. In the case of randomized trials, the following data were extracted: sample size (randomized and non-randomized), reasons for non-randomization, exclusions after randomization, drop-outs, and ‘intention-to-treat’ analysis. For all studies, available information on inclusion and exclusion criteria, mono- or multicenter study design, primary and secondary outcome measures, and surgical characteristics such as type of surgery and complications were registered. Furthermore, general descriptive data (such as sex, age, body mass index (BMI), and American Society of Anesthesiology (ASA) classification) were evaluated and presented in an additional table. Extracted data were stored and managed using the review manager software Covidence [31].

#### 2.3.3. Assessment of Risk of Bias in Included Studies

The methodological quality of RCTs was assessed using the Cochrane risk-of-bias tool (original (2016) version), RoB 2 [32]. The quality of observational studies was assessed using the ROBINS-I tool (2019 version) [33]. Every study was assessed independently by EG, OM, and MS. Discrepancies were solved by consensus discussion with a fourth reviewer, JS.

#### 2.3.4. Measures of Treatment Effect

A descriptive quantitative analysis of the included quantitative studies was performed for continuous primary outcomes (participation rate) and secondary outcomes. For the qualitative data, a thematic synthesis was applied by identifying common themes and subthemes across the studies. An inductive approach was used. Subsequently, data were organized and categorized in a thematic framework.

#### 2.3.5. Dealing with Missing Data

When data were missing, we investigated whether this data were missing at random using the Little MCAR test [34], where indicated, combined with multiple imputation [35]. Depending on the outcome, missing data were either considered to have an impact on the outcome or not.

#### 2.3.6. Handling Duplicate and Companion Publications

In cases of duplicate publications, companion papers, or multiple reports of a primary trial, the information was consolidated by collecting and assessing the available data, using the most comprehensive data set across all known publications. Excluded overlapping articles were checked for relevant information regarding outcome measures. If there was minimal overlap, both studies were included. Multiple reports of primary trials were cited as secondary references under the study ID of the included trial.

#### 2.3.7. Assessment of Heterogeneity

Heterogeneity across studies was anticipated due to variations in inclusion and exclusion criteria, prehabilitation program components (e.g., nutritional, physical, psychological), duration, and outcome measures. To assess heterogeneity, we compared the fixed-effect and random-effect estimates of the intervention effect. In case of no discrepancy and no heterogeneity, the fixed-effect models were presented. In case of discrepancy between the two models, both results were reported.

In case of considerable heterogeneity (>75%) [27], descriptive quantitative analysis and qualitative analysis were performed, and the outcomes of included studies were described.

#### 2.3.8. Assessment of the Body of Evidence

The body of evidence was assessed by using the GRADE approach [36,37]. Four descriptors (‘very low’, ‘low’ ‘moderate’, and ‘high’) were used to indicate the confidence level in the evidence.

## 3. Results

### 3.1. Results of the Search

In the initial search, performed in October 2023, the systematic search identified 35,819 articles. The literature search and selection processes are shown in Figure 1. After removing the duplicates, 19,646 references were screened and assessed for eligibility based on title and abstract. Of these, 99 full-text articles were assessed for eligibility, of which 85 studies met the inclusion criteria. To incorporate relevant articles, that were recently published, the search was repeated in December 2024. This second search identified 2455 articles. Based on the screening of the title and abstract, 31 articles were subsequently assessed by full-text reading. Ten extra studies were included. No additional studies were included based on the reference checks. In total, 95 articles met the inclusion criteria. After excluding six studies due to the full overlap of study samples [38,39,40,41,42,43], a number of 89 studies were included in the final analysis.

### 3.2. Included Studies

#### 3.2.1. Study Design and Setting

This systematic review included 34 randomized controlled trials [1,18,44,45,46,47,48,49,50,51,52,53,54,55,56,57,58,59,60,61,62,63,64,65,66,67,68,69,70,71,72,73,74,75] and 55 [15,23,24,26,76,77,78,79,80,81,82,83,84,85,86,87,88,89,90,91,92,93,94,95,96,97,98,99,100,101,102,103,104,105,106,107,108,109,110,111,112,113,114,115,116,117,118,119,120,121,122,123,124,125,126] observational studies. Of the 55 observational studies, 6 articles had a qualitative study design [24,122,123,124,125,126]. Furthermore, most cohort studies were of a prospective nature.

The included studies involved a total number of 13,383 patients: 7162 patients in the prehabilitation cohort and 6272 in the control cohort. The total number of study participants ranged from 10 to 761 (median 75, IQR 29–191), with a range of 0–379 patients in the prehabilitation cohort (median 45, IQR 21–102) and 0–382 in the control cohort (median 24, IQR 0–80). Three studies (3.4%) reported a number of zero participants in the prehabilitation cohort [23,122,124] after recruitment and enrollment.

The median length of follow-up after surgery was 56 days [15,23,26,30,31,32,33,34,35,36,37,38,39,40,41,42,43,44,45,46,47,48,49,50,51,52,53,54,55,56,57,58,59,60,61,62,63,64,65,66,67,68,69,70,71,72,73,74,75,76,77,78,79,80,81,82,83,84,85,86,87,88,89,90,91,92,93,94,95,96,97,98,99,100,101,102,103,104,105,106,107,108,109,110,111,112,113,114,115,116,117]. Studies were performed between 1995 and 2022. Most studies were conducted in Western countries such as Canada, the Netherlands, and the United Kingdom.

#### 3.2.2. Participants

Regarding inclusion criteria, 69 studies included CRC patients only [1,15,18,23,26,43,46,47,48,49,50,51,52,53,54,55,56,57,58,59,60,61,62,63,64,65,68,69,70,71,72,73,74,75,76,77,78,79,80,81,83,84,85,88,89,93,94,97,98,99,101,102,104,105,106,107,108,109,110,113,114,117,119,121,122,123,124,125,126]. Other studies involved CRC patients and also patients undergoing prehabilitation for upper gastrointestinal (*n* = 7) [44,45,87,91,95,112,115], hepato-pancreatic-biliary (*n* = 4) [44,87,95,112], lung (*n* = 3) [24,87,96], urological (*n* = 3) [95,100,116] or other surgeries (*n* = 10) [66,67,82,87,90,92,103,111,118,120]. The median number of patients in the prehabilitation cohort with colon cancer was 26 [9,10,11,12,13,14,15,16,17,18,19,20,21,22,23,24,25,26,27,28,29,30,31,32,33,34,35,36,37,38,39,40,41,42,43,44,45,46,47], with a median percentage of 65.7% [29.9–72.1%]. The median number of patients with rectal cancer was 18 [6,7,8,9,10,11,12,13,14,15,16,17,18,19,20,21,22,23,24,25], with a median percentage of 32.0% [15.9–45.6%].

The control group included a median number of 24 participants [8,9,10,11,12,13,14,15,16,17,18,19,20,21,22,23,24,25,26,27,28,29,30,31,32,33,34,35,36,37,38,39,40,41,42,43,44,45,46,47,48,49,50,51,52,53,54,55,56,57,58,59,60,61,62,63,64,65,66,67,68,69,70,71,72,73,74,75,76,77,78,79,80,81,82,83,84,85,86,87,88,89,90,91,92,93,94,95,96,97,98,99,100,101,102,103] with colon cancer, at a median percentage of 61.0% [39.5–76.1%], and 17 [6,7,8,9,10,11,12,13,14,15,16,17,18,19,20,21,22,23,24,25,26] participants with rectal cancer, at a median percentage of 30.5% [18.0–47.8].

Sixty-eight studies included only patients undergoing oncological surgery [1,15,23,24,26,43,45,47,48,49,51,52,53,54,56,57,59,61,62,63,64,65,68,69,70,71,72,73,74,75,76,77,78,79,80,83,84,85,88,89,90,92,93,94,95,96,97,98,99,100,101,102,106,107,108,109,110,112,113,114,115,116,117,119,120,121,122,123,126], and six studies included both benign and oncological surgery [18,44,46,50,58,60]. Nine studies included patients receiving neoadjuvant chemotherapy [23,26,62,64,70,79,89,102,112]. In the prehabilitation cohort, the majority of patients underwent minimally invasive surgery, with a median of 80.7% [53.1–95.6]. In comparison, 11.9% [0.9–38.7] of patients underwent open surgery. Minimally invasive surgery was also most prevalent in the control group, with a median of 74.4% [40.0–93.0], versus 19.5% [1.6–47.1%] of patients receiving open surgery.

Regarding the exclusion criteria, emergency or non-elective surgery was an exclusion criterion for 22 studies [1,43,44,45,48,54,60,63,65,74,75,78,80,81,85,92,93,94,108,114,117,126], whereas 51 studies excluded patients based on limited physical or psychiatric functioning [1,23,26,43,44,45,46,48,50,51,53,54,55,56,57,58,59,60,61,63,64,65,66,68,69,73,74,75,77,79,80,82,85,87,88,89,92,93,98,100,101,108,110,113,115,116,117,118,119,125,126]. An overview of inclusion and exclusion criteria is provided in Appendix A, and patient characteristics are displayed in Appendix A.

### 3.3. Interventions and Comparisons

#### Prehabilitation Programs

Upon analyzing the prehabilitation programs, we noticed differences in prehabilitation modalities to improve patients’ condition before surgery. The majority of studies (*n* = 50, 56.2%) involved a multimodal prehabilitation program, with exercise or nutritional intervention. Twenty-seven studies (30.3%) described a unimodal exercise-based prehabilitation program, 10 studies (11.2%) described a unimodal nutrition-based prehabilitation program, and two studies (2.2%) involved a preoperative nutritional intervention and a (postoperative) rehabilitation program. An overview of the prehabilitation interventions is provided in Appendix A.

Due to the large heterogeneity in prehabilitation programs, no meta-analysis could be performed. This heterogeneity arises from variations in inclusion and exclusion criteria, as well as differences in the length, content, and outcomes of the prehabilitation programs. Please refer to Appendix A. The information was insufficient to conclude certainty of evidence.

### 3.4. Primary Outcome—Quantitative Data

Of the sixty-three studies describing participation, the participation rate ranged from 0 to 99.4%. The median rate of participation in the studies was 66.6% [55.6–83.6%]. Reasons for declining participation in the study were reported in 38 studies (42.7%). The most common reasons for declining participation were logistical factors, such as lack of transport to the training location and being too busy, as well as medical reasons, including contraindications for training due to arthrosis or cardiopulmonary failure. Other common reasons included a lack of interest in prehabilitation, the perceived burden of the prehabilitation program, concerns about a potential delay in surgery, or no need to prehabilitate (self-perceived), this is displayed in Figure 2.

#### 3.4.1. Participation—Unimodal Prehabilitation

In total, 27 studies (30.3%) involved a unimodal exercise prehabilitation program, including a total number of 1870 patients with 1058 patients in the prehabilitation cohort. The number of participants in the prehabilitation program ranged from 0 to 137, because two feasibility trials [23,122] (7.4%) could not include participants in their final data analysis. The median participation rate was 61.0% [41.5–73.6%]. Of these studies, 15 (55.6%) provided information about the participation rate or the reasons for agreeing/declining to participate. The most common reasons for declining participation were a busy schedule and logistical issues.

Ten studies (11.2%) involved a program with nutritional intervention, with 619 patients included in the prehabilitation cohort. The number of participants ranged from 20 to 132, with a median of 59 participants in the prehabilitation cohort [46,47,48,49,50,51,52,53,54,55,56,57,58,59,60,61,62,63,64,65,66,67,68,69]. Five studies (50%) described participation rates, with a median participation rate of 77.6% [IQR 53.9–90.6%]. In the four studies describing the reason for declining participation, the most common reason was the refusal of the nutritional intervention.

#### 3.4.2. Participation—Multimodal Prehabilitation

Fifty studies (56.2%) described a multimodal prehabilitation program of which 36 (72%) provided information about participation. The range of participants was 5–641 with a total of 5413 participants in the prehabilitation cohort. The median participation rate was 70.0% [57.1–84.8]. Twenty-one studies reported reasons for not participating of which logistical reasons and the lack of interest were most common.

#### 3.4.3. Participation—Exercise Intervention (*n* = 74)

When distinguishing between exercise and nutritional interventions, 57 studies involving exercise prehabilitation described participation rates (77.0%). The number of participants in the prehabilitation cohort varied from 0 to 641, with a total of 6258 patients included in the exercise prehabilitation cohort. The median participation rate was 67.1% [53.7–83.4%].

#### 3.4.4. Participation—Nutritional Intervention (*n* = 57)

Thirty-seven nutritional prehabilitation studies described participation (64.9%). The number of participants varied from 0 to 641, with a total of 5073 patients included. The median participation rate was 75.0% [56.3–84.6%].

#### 3.4.5. Adherence

The length of prehabilitation ranged from 7 to 115 days. Of studies describing mean prehabilitation duration, the mean was 28.2 days (95% CI 18,2–38.1, SD 22.4). Studies reporting a median prehabilitation duration had an overall median of 33 days [21,22,23,24,25,26,27,28,29,30,31,32,33,34,35,36,37,38,39,40]. Overall, 70 studies (78.7%) provided information on adherence to prehabilitation programs. The adherence rates ranged from 16 to 100%. Of studies reporting overall adherence (*n* = 14, 15.7%), the median was 72% [58.8–82.3%].

Reasons for non-adherence were reported by 38 studies, the most common reasons for non-adherence were medical reasons such as physical injuries and logistical issues (See Figure 3).

#### 3.4.6. Adherence—Exercise Intervention (*n* = 74)

Seventy-one studies involving exercise prehabilitation (95.9%) described the method of adherence measurement during exercise interventions. In most studies program adherence was reported by the supervisor (*n* = 23, 32.4%), or by both the patients and supervisors (*n* = 26, 36.6%) Less often patients reported their activity themselves (*n* = 20, 28.2%), for example by using an activity log.

Fifty-nine studies provided information on adherence to exercising (79.7%), with a median of 75.4% compliance rate [62.0–86.0%].

For unimodal exercise prehabilitation the median adherence rate was 78.0% [66.1–90.8%] in the 24 of 27 (88.9%) studies. Multimodal exercise prehabilitation had a median adherence rate of 72.0% [61.0–84.0%] in 35 of 47 studies describing multimodal exercise prehabilitation.

#### 3.4.7. Adherence—Nutritional Intervention (*n* = 57)

The method of measuring adherence with nutritional interventions was described in 33 studies (57.9%) of 57 studies involving nutritional prehabilitation. Adherence was most often measured by self-reporting for most patients (75.8%), for example, in a food diary. Supervision was less common (12.1%), followed by 9.1% of a combination of self-reporting and supervision. Twenty-two studies provided information on the compliance rate to nutritional interventions (38.6%). The median compliance rate was 77.0% [65.4–96.3%].

The median adherence rate was 78.0% [66.1–90.8%] in the unimodal nutritional interventions (*n* = 24, 88.9%). Multimodal prehabilitation involving nutritional interventions was adhered to by 66.7% of patients (median, IQR [57.8–95.3%]).

### 3.5. Primary Outcome—Qualitative Data (n = 6)

Three studies involved a multimodal prehabilitation program [24,125,126], one study a unimodal exercise program [122], and two studies [123,124] interviewed CRC patients about their visions towards prehabilitation. The number of participants in the prehabilitation cohort ranged from 0 to 34, with a median participation rate of 80%.

#### 3.5.1. Participation

The most common enablers and barriers to participation are displayed in Figure 4.

#### 3.5.2. Adherence

In two studies, adherence to exercise interventions was measured by a supervisor [24,125]. The adherence to nutritional interventions was self-reported in two studies [125,126].

Figure 5 displays the most important reported enablers and barriers to adherence with prehabilitation.

### 3.6. Risk of Bias

The methodological quality of the included quantitative studies (*n* = 83) is presented in risk of bias (RoB) summaries (Figure 6, Figure 7, Figure 8 and Figure 9). Fourteen studies [43,55,66,78,91,95,97,99,101,103,106,110,115,120] (16.8%) were assessed as having a serious or critical risk of bias, primarily due to confounding, selection bias, bias in measurement or bias in the selection of reported results. Sixty-one studies [15,23,26,44,45,46,47,48,50,52,56,57,58,59,60,61,62,63,64,65,67,68,69,71,72,73,75,76,77,79,80,81,82,83,84,85,87,88,89,90,92,93,94,96,98,100,102,104,105,107,108,109,111,112,113,114,116,117,118,119,121] (73.5%) were deemed to have a moderate risk of bias. Eight studies [1,18,49,51,53,54,70,74] (9.6%) were considered to have a low risk of bias. Overall, the methodological quality of the included studies ranged from moderate to low.

An overview of the quality of the included qualitative studies is provided in Appendix A. One study [24] had a low trustworthiness, the other five studies [122,123,124,125,126] had a medium or high trustworthiness.

## 4. Discussion

This systematic review identified and analyzed 89 studies investigating prehabilitation programs for patients undergoing colorectal cancer surgery. It is the first review to collect qualitative and quantitative data to examine the current literature on participation and adherence to prehabilitation programs. Most of the included studies were prospective cohort studies. The results of the studied sample (*n* = 13,383) demonstrated a large variety in the participation rate (0–99.4%). Data on recruitment and declining participation was limited. The available data displayed that most patients eligible for inclusion were recruited. Participation was most often refused based on logistical or medical reasons. The highest participation rates were described for unimodal nutritional prehabilitation.

Adherence also varied widely among studies, with an adherence rate of 16–100%. A minority of studies described reasons for (non-) adherence. When reported, adherence was most often hindered due to medical reasons. The quantitative and qualitative results emphasize the importance of addressing accessibility for patients with medical limitations and providing help for logistical issues. Also, since high adherence is crucial for successful prehabilitation [51,127], more attention is paid to describing the measurement of adherence and the adherence rates to generate more information on how to improve adherence. The analyzed articles suggest that professional guidance, peer support, and supportive tools and content could improve adherence.

### 4.1. Participation in Prehabilitation Programs

The recruitment and enrollment of patients into clinical studies have long been recognized as challenging, with important implications for study outcomes, especially for prehabilitation programs, where the most motivated patients are also most likely to be adherent and achieve the best results [128].

In general, approximately 50% of clinical trials are estimated to fail to meet their recruitment targets [129]. Our review reflects this difficulty, with a reported average participation rate of 66.6%. This finding is consistent with the meta-analysis of Xu et al. [130]. However, this figure is relatively high compared to some studies such as the pooled retrospective analysis of Lee et al. [131]. The difference in participation rate could be attributed to the difference in patient population, including patients undergoing surgery for lung, and esophageal cancer. In contrast, Cuijpers et al. [132] reported a wide range in participation rates, which aligns with our findings underscoring the heterogeneity in patient recruitment and engagement across prehabilitation studies.

### 4.2. Participation—Barriers

Logistical barriers, such as transportation difficulties and scheduling conflicts, emerged as primary obstacles to participation in prehabilitation programs. These barriers align with the findings of the qualitative evidence synthesis of Houghton et al. [129], who reported that additional appointments or travel requirements can be burdensome for patients. Medical contraindications, including comorbidities and perceived ineligibility, along with a perception of surgery urgency, were reported as reasons for non-participation in prehabilitation programs. Similarly, the systematic reviews of Alsuwaylihi et al. [133] and Cuijpers et al. [132] highlighted psychological barriers, such as feeling overwhelmed by the number of prehabilitation sessions or preferring immediate surgery, as significant deterrents. Furthermore, inadequate communication, lack of specific information, and an over-emphasis on benefits were reported as additional barriers. Although this study did not collect data on socio-economic status, qualitative findings suggest that financial burden could limit participation. This aligns with the findings of Lee et al. [131] who found that individuals from lower socio-economic backgrounds were less likely to engage in prehabilitation programs, probably due to financial limitations and disparities in healthcare access.

### 4.3. Participation—Enablers

On the other hand, several factors were identified as enablers of higher participation rates in prehabilitation programs. The qualitative findings of this review highlight both psychological and physical benefits, such as creating a sense of regained control, a positive mindset, reduced stress, and the opportunity to optimize physical fitness to withstand surgical stress better.

Clear and direct communication from healthcare providers (HCPs) was a key facilitator for participation, as was the provision of detailed information about the prehabilitation program. Support from both relatives and HCPs played a crucial role in encouraging participation. These findings align with Houghton et al. [129], who reported that clear, face-to-face communication from HCPs enhanced patient understanding of the prehabilitation process and improved participation in prehabilitation programs. Similarly, family support and encouragement from HCPs were found to be influential in motivating patients to participate in prehabilitation programs. These findings highlight the importance of effective communication and social support for improving patient engagement in prehabilitation programs.

Notably, this review showed the highest recruitment rate for unimodal nutritional intervention, suggesting that patients perceive this as less burdensome than multimodal prehabilitation programs.

### 4.4. Adherence to Prehabilitation Protocols

Adherence to prehabilitation programs is essential for an accurate evaluation of the effectiveness of these interventions. Adherence to long-term therapies has been reported to average around 50% [134], and given that prehabilitation typically involves a multifaceted approach, adherence may be even more challenging. The average adherence rate of this review is consistent with findings by Alsuwaylihi et al. [133] and O’Doherty et al. [135]. Similarly, Luther et al. [136] emphasized the heterogeneity in adherence rates, particularly in studies involving exercise interventions, with adherence varying widely across different study populations.

### 4.5. Adherence—Barriers

This review reported medical reasons, rescheduled surgery, and logistical issues as the most important barriers to adherence. These findings are consistent with the review of O’Doherty et al. [135] and Alsuwaylihi et al. [133], describing reduced adherence when patients had the opportunity to undergo surgery earlier. Competing life responsibilities, such as work and family obligations, were also cited as reasons for non-adherence, as well as medical issues and fatigue. Additionally, logistical barriers, such as transportation difficulties and the need for frequent appointments, were common challenges that reduced adherence to prehabilitation interventions [132,135].

Cuijpers et al. [132] reported that side effects from neoadjuvant chemoradiotherapy (NACRT) were also a barrier to adherence. Patients undergoing NACRT reported feeling unwell or fatigued, which negatively impacted their ability to participate in prehabilitation programs. Personal issues, such as stress or family responsibilities were also frequently reported as reasons for non-adherence.

### 4.6. Adherence—Enablers

Several factors were identified as contributing to higher adherence rates. This review highlights the positive impact of professional guidance, flexible and patient-centered programs, peer support, and supportive tools and content. These findings align with previous research, describing that a holistic, patient-centered approach that integrates multiple aspects of care contributed significantly to improved adherence [137,138]. Consistent with this review, the involvement of a multidisciplinary team also was a key enabler of adherence, as this approach provides comprehensive care and fosters a supportive environment for patients. Le Roy et al. [139] noted that such teams contribute to higher patient adherence by addressing the diverse needs of patients, from exercise to psychological support. Furthermore, the early integration of prehabilitation into the treatment process was identified as an important factor, as it builds trust between patients and healthcare providers, which can enhance patient engagement and reduce dropouts [140].

The highest adherence was reported for unimodal exercise or nutritional prehabilitation, suggesting that multimodality prehabilitation is more challenging for patients to adhere to.

In contrast, Alsuwaylihi et al. [133] established that multimodal prehabilitation programs (which combined exercise, nutrition, and psychological support) were associated with higher adherence compared to unimodal interventions. Additionally, longer prehabilitation programs (≥3 weeks) and home-based training interventions were linked to better adherence, suggesting that the flexibility of the program and the opportunity to progress into prehabilitation may facilitate higher adherence.

### 4.7. Strengths and Limitations of the Review

The strengths of this systematic review are the large number of studies included and the diversity of study designs, which enhance the generalizability of the findings. By integrating qualitative and quantitative data, this review provides a comprehensive understanding of the factors influencing patient participation and adherence in prehabilitation programs.

However, several limitations must be acknowledged before the generalization of findings. We found the overall methodological quality of the studies about prehabilitation programs in colorectal cancer to be of medium to low quality. Nonetheless, all studies were included to provide a thorough overview of the current literature. Several studies lacked detailed information regarding participation rates. Also, both adherence measurement and adherence data are missing in various studies which hampers generalizability. Moreover, no meta-analysis could be conducted due to the large heterogeneity of study designs, intervention protocols, and outcome measures. This impairs direct comparisons between the studies. Additionally, the predominance of studies conducted in Western countries limits the applicability of these findings to non-Western healthcare settings, where cultural, socio-economic, and healthcare system differences may impact participation as well as the outcomes of prehabilitation programs. On the other hand, these countries have facilities to offer prehabilitation and are most likely to integrate it into their care pathway.

Moreover, the observational nature of many of the included studies introduces the potential for confounding, which could influence the reported outcomes. The low number of randomized controlled trials on the topic of prehabilitation in colorectal cancer underscores the need for further high-quality research to provide robust evidence for prehabilitation interventions. The limited number of studies reporting on participation and adherence rates further hinders the ability to draw definitive conclusions about the factors influencing these outcomes.

The lack of detailed information on patients who declined participation or did not adhere to the intervention introduces a risk of bias in outcome measurement and weakens the strength of evidence. Additionally, the absence of detailed descriptions of control group interventions in several studies may act as a potential confounder.

### 4.8. Recommendations for Future Research

Future research should focus on developing standardized protocols for measuring participation and adherence to enhance the comparability of results across studies. Furthermore, further studies should explore strategies to improve participation and adherence, particularly in home-based or digitally supported prehabilitation interventions. Large-scale randomized controlled trials or stepped-wedge design studies are necessary to identify the most effective and feasible prehabilitation approaches, especially in diverse healthcare settings. Additionally, research should examine the experiences of patients who decline participation in prehabilitation programs to better understand the barriers to engagement from a broader perspective.

## 5. Conclusions

This systematic review provides a comprehensive overview of the current literature on prehabilitation programs for patients undergoing colorectal cancer surgery. Despite the postulated beneficial impact of prehabilitation on physical and psychological status, significant variability in study designs, participation rates, and intervention strategies suggests that further research is needed to identify the most effective prehabilitation approaches. Addressing logistical barriers and patient concerns, particularly through flexible, patient-centered interventions, may enhance both participation and adherence, maximizing the benefits of prehabilitation in clinical practice. Future studies should focus on large-scale RCTs, explore diverse healthcare settings, and compare strategies to improve both participation and adherence to prehabilitation programs.

## Figures and Tables

**Figure 1 nutrients-17-01792-f001:**
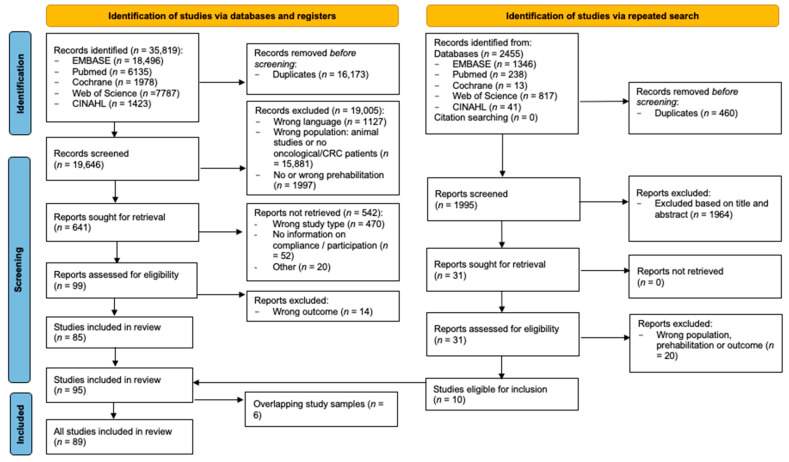
PRISMA flowchart.

**Figure 2 nutrients-17-01792-f002:**
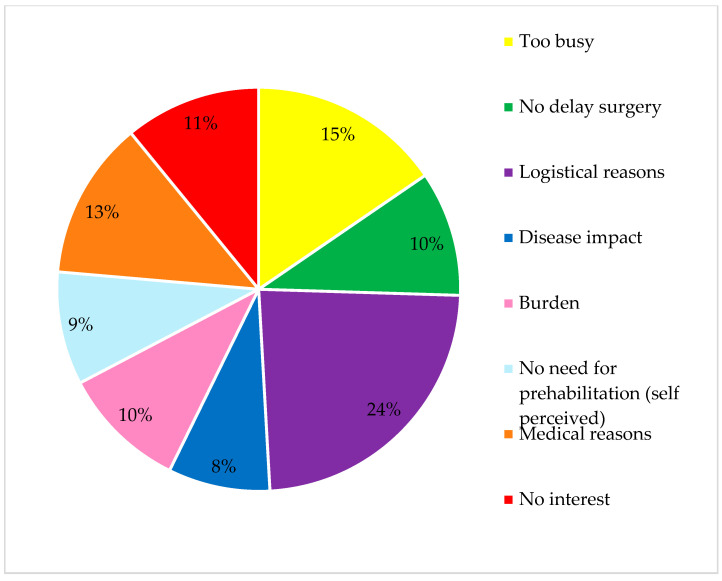
Reasons for declining participation.

**Figure 3 nutrients-17-01792-f003:**
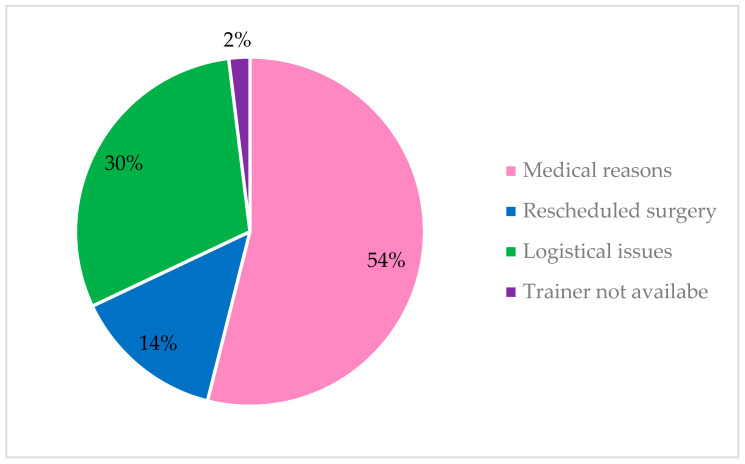
Reasons for non-adherence.

**Figure 4 nutrients-17-01792-f004:**
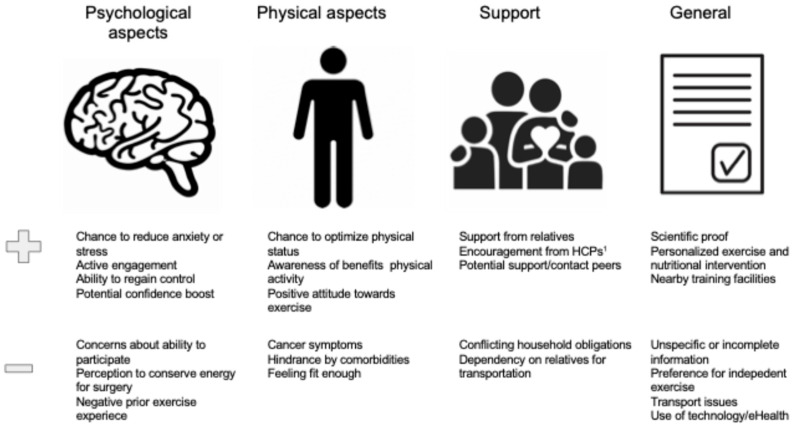
Enablers and barriers to participation: qualitative studies. ^1^ HCP, Healthcare professional.

**Figure 5 nutrients-17-01792-f005:**
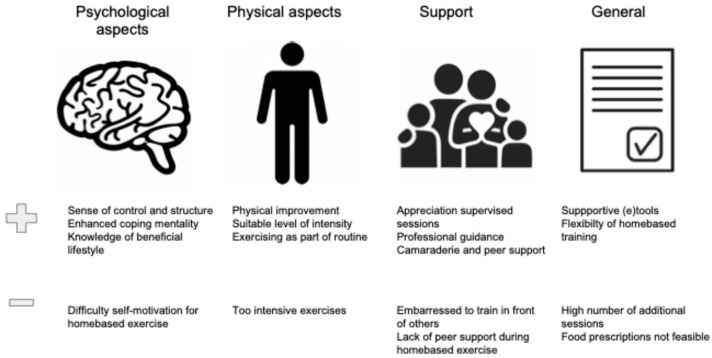
Enablers and barriers to adherence: qualitative studies.

**Figure 6 nutrients-17-01792-f006:**
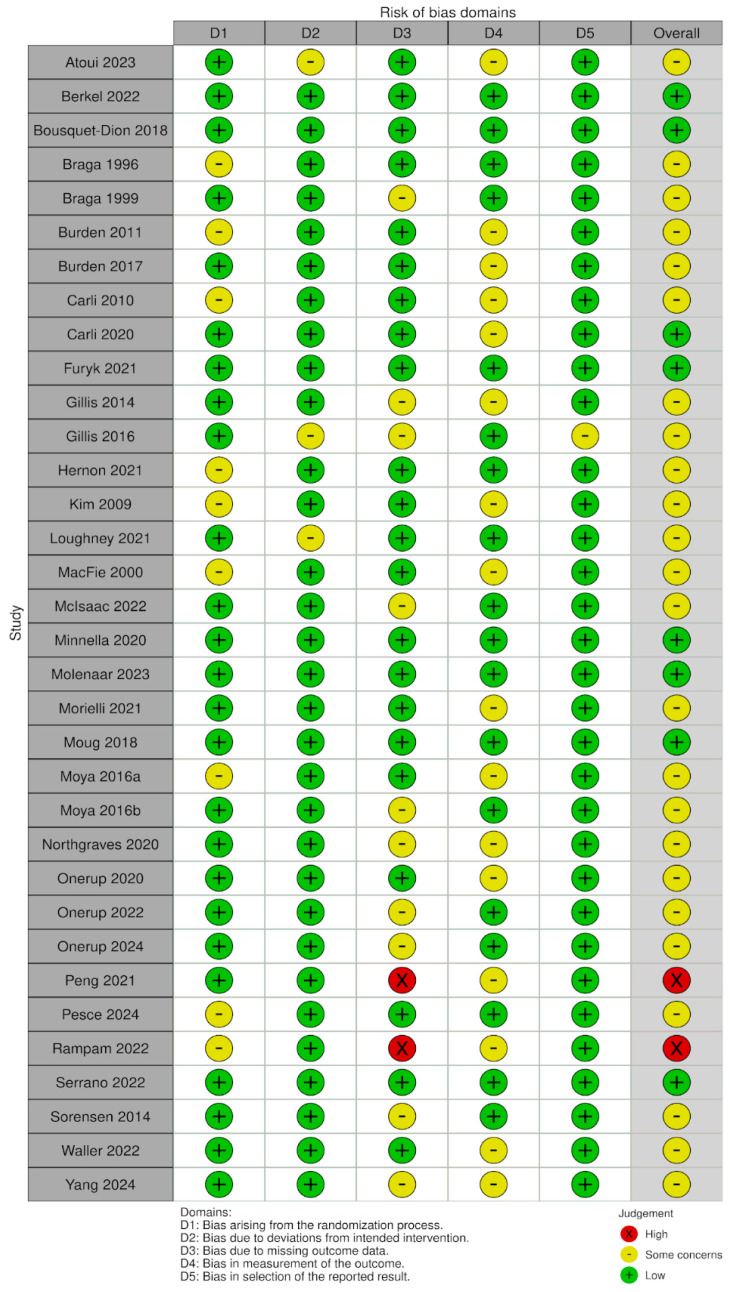
RoB-2 plot risk of bias [1,18,44,45,46,47,48,49,50,51,52,53,54,55,56,57,58,59,60,61,62,63,64,65,66,67,68,69,70,71,72,73,74,75].

**Figure 7 nutrients-17-01792-f007:**
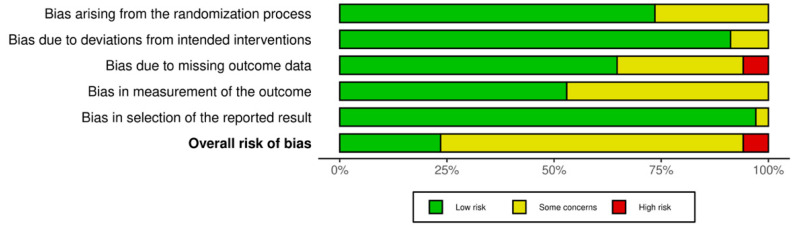
RoB 2 Weighted summary plot.

**Figure 8 nutrients-17-01792-f008:**
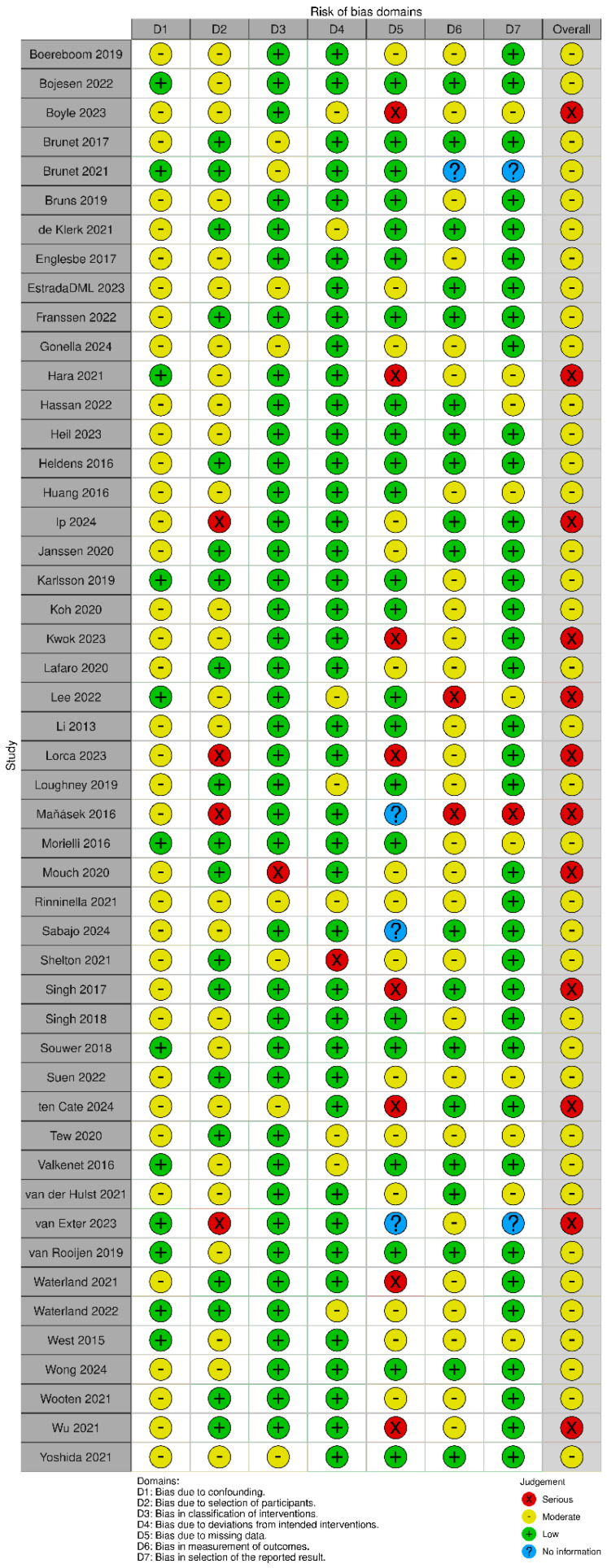
ROBIN-I plots risk of bias [15,23,24,26,76,77,78,79,80,81,82,83,84,85,86,87,88,89,90,91,92,93,94,95,96,97,98,99,100,101,102,103,104,105,106,107,108,109,110,111,112,113,114,115,116,117,118,119,120,121,122,123,124,125,126].

**Figure 9 nutrients-17-01792-f009:**
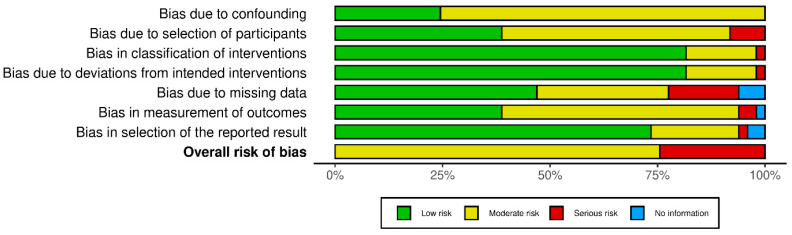
ROBINS-I weighted summary plot.

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
