# Peer review of "Participation and Adherence to Prehabilitation Programs for Colorectal Cancer"

_nutrients, 2025, doi:10.3390/nu17111792_

Round 1
Reviewer 1 Report
Comments and Suggestions for Authors
Authors reviewed the articles to “summarize all reported cases of common peroneal nerve paralysis after weight loss,” (L19). However, this article has not fully answered some of the questions due to insufficient description.
First, authors suggest “Due to the large heterogeneity in prehabilitation programs” (L247), but they did not explain “the large heterogeneity”. Although authors suggest “due to the large heterogeneity of study designs, intervention protocols, and outcome measures” (L491) in limitation section, they should justify it before this sentence in method section. Authors should add descriptions regarding “the large heterogeneity” in result section.
SEcond, authors described some of sentences without citation or justification as follows; “Functional capacity has been identified as a key determinant for a favorable outcome after surgery. As a result, there is a strong impetus for clinicians to improve patients’ functional capacity.” (L43), “the little MCAR test” (L153), and “two studies” (L337), but it is difficult for readers to judge them without references as evidence for each description. Authors should add references for these descriptions.
Minor comment
L63: “This study aims to explore how to improve patients’ participation in and adherence to prehabilitation programs” may be “This study aims to explore how to improve patients’ participation in and adherence to prehabilitation programs for colorectal cancer”.
L117: “The titles, abstracts, and descriptor terms of all downloaded material from the electronic searches were read by MS, all irrelevant reports were excluded. All citations identified were independently assessed by MS and OM to determine the relevance of each article based on the pre-specified criteria.” may be “The titles, abstracts, and descriptor terms of all downloaded material from the electronic searches were read by one researcher (MS), all irrelevant reports were excluded. All citations identified were independently assessed by two researchers (MS and OM) to determine the relevance of each article based on the pre-specified criteria.”.
L178: The sentence “All references from the four search databases were imported into Rayyan bibliographic software[33].” should be in method section.
L179: “The literature search and selection processes are shown in Figure 1[26].” may be “The literature search and selection processes are shown in Figure 1.”, and the use of PRISMA flowchart should be explained in method section with citation of reference number 26.
Author Response
Reviewer 1
Comment 1: Authors reviewed the articles to “summarize all reported cases of common peroneal nerve paralysis after weight loss,” (L19). However, this article has not fully answered some of the questions due to insufficient description.
Response 1: We would like to thank the reviewer for their time and feedback. However, we believe this comment may have been submitted in error, as our manuscript does not focus on the topic of peroneal nerve paralysis after weight loss. Please kindly advise if this comment was intended for a different manuscript.
Comment 2: First, authors suggest “Due to the large heterogeneity in prehabilitation programs” (L247), but they did not explain “the large heterogeneity”. Although authors suggest “due to the large heterogeneity of study designs, intervention protocols, and outcome measures” (L491) in limitation section, they should justify it before this sentence in method section. Authors should add descriptions regarding “the large heterogeneity” in result section.
Response 2: We thank the reviewer for this helpful suggestion. We describe our approach to heterogeneity assessment in L172-177 of the Methods section. In addition, we added a sentence on expected heterogeneity in L173-175: “Heterogeneity across studies was anticipated due to variations in inclusion and exclusion criteria, prehabilitation program components (e.g., nutritional, physical, psychological), duration, and outcome measures.”
In the Results, we added a clarifying sentence in L258-260. With this sentence we aim to clarify and support our statement on the variability across the included studies: “This heterogeneity arises from variations in inclusion and exclusion criteria, as well as differences in the length, content, and outcomes of the prehabilitation programs. Please refer to appendices A, C, and D.” The Appendices A, C and D, which provide structured overviews of the study populations and program components.
Comment 3: Second, authors described some of sentences without citation or justification as follows; “Functional capacity has been identified as a key determinant for a favorable outcome after surgery. As a result, there is a strong impetus for clinicians to improve patients’ functional capacity.” (L43), “the little MCAR test” (L153), and “two studies” (L337), but it is difficult for readers to judge them without references as evidence for each description. Authors should add references for these descriptions.
Response 3: We sincerely value the insightful feedback from the reviewer. It is essential to add references to ensure the manuscript is transparent and comprehensible for readers. We therefore inserted the matching references to our manuscript. Please see L43-45, L162, and L353-355.
Minor comments
Comment 4: L63: “This study aims to explore how to improve patients’ participation in and adherence to prehabilitation programs” may be “This study aims to explore how to improve patients’ participation in and adherence to prehabilitation programs for colorectal cancer”.
Response 4: We are thankful for the reviewer's suggestion. We agree on the value of adding “for colorectal cancer” to provide essential detail on the aim of the systematic review and we added it to L69.
Comment 5: L117: “The titles, abstracts, and descriptor terms of all downloaded material from the electronic searches were read by MS, all irrelevant reports were excluded. All citations identified were independently assessed by MS and OM to determine the relevance of each article based on the pre-specified criteria.” may be “The titles, abstracts, and descriptor terms of all downloaded material from the electronic searches were read by one researcher (MS), all irrelevant reports were excluded. All citations identified were independently assessed by two researchers (MS and OM) to determine the relevance of each article based on the pre-specified criteria.”.
Response 5: We are delighted with the constructive feedback from the reviewer. We have integrated the suggestion made into the manuscript, please see L124-127.
Comment 6: L178: The sentence “All references from the four search databases were imported into Rayyan bibliographic software[33].” should be in method section.
Response 6: We want to thank the reviewer for this suggestion. We have moved the content from the results section to the methods section, as it is more appropriate there. The sentence has been removed from the results section and inserted into the methods section at L118.
Comment 7: L179: “The literature search and selection processes are shown in Figure 1[26].” may be “The literature search and selection processes are shown in Figure 1.”, and the use of PRISMA flowchart should be explained in method section with citation of reference number 26.
Response 7: We are grateful for the attentive suggestion of the reviewer. We added a description on the use of the flowchart and PRISMA guideline in the Methods in L133 and we have removed the reference in line 190.

Reviewer 2 Report
Comments and Suggestions for Authors
Our fellow researchers conducted a study that led to this paper, whose aim is the preoperative improvement of patients' functional capacity—a goal summarized by the appropriate term prehabilitation. The abstract provides a good summary of the entire work. The introduction clearly outlines the difficulties faced by healthcare professionals in implementing prehabilitation programs in patients with colorectal cancer. I would add that, at least in Western countries, screening programs have often reduced both the number of complicated cases and the total number of colorectal cancer patients, as preventive colonoscopies allow for the removal of polyps through polypectomies. This screening effort has led to a colorectal cancer population that is generally older and already affected by reduced motor and/or cognitive abilities, making the implementation of prehabilitation programs more challenging. Hospitals should be equipped with dedicated personnel, or ideally, physiotherapy facilities should be available near hospitals. Since this paper is essentially a meta-analysis with significant biases related to the heterogeneity of the studies considered, the inclusion and exclusion criteria of the articles selected from international literature are clearly explained. The study selection process was conducted using well-described and fully reproducible criteria for any group interested in replicating the work. The results are well described and highlight two key pillars: what we could call preoperative physiotherapy, and the involvement of the nutritionist (see doi.org/10.3390/nu17010188, which should be read and cited in the bibliography), since all oncology patients should be considered "de facto" malnourished or, more accurately, undernourished. In fact, we strongly recommend stating that, during the multidisciplinary team discussion after diagnostic workup and before determining the appropriate therapeutic pathway, the patient should also be assessed by a nutritionist to improve their condition ahead of any medical or surgical treatment. In the discussion section, we appreciated the inclusion of family members and caregivers, as well as the barriers encountered by researchers, which ultimately translate into the paper's biases. Psychological support is an excellent addition to prevent dropouts or non-adherence to programs. Congratulations on this first paper on the subject. As a final recommendation, consider shortening it slightly, as some parts—particularly in the results and discussion—are somewhat redundant. Excellent visuals, solid English, and an outstanding bibliography.
Author Response
Reviewer 2
Comment: Our fellow researchers conducted a study that led to this paper, whose aim is the preoperative improvement of patients' functional capacity—a goal summarized by the appropriate term prehabilitation. The abstract provides a good summary of the entire work. The introduction clearly outlines the difficulties faced by healthcare professionals in implementing prehabilitation programs in patients with colorectal cancer. I would add that, at least in Western countries, screening programs have often reduced both the number of complicated cases and the total number of colorectal cancer patients, as preventive colonoscopies allow for the removal of polyps through polypectomies. This screening effort has led to a colorectal cancer population that is generally older and already affected by reduced motor and/or cognitive abilities, making the implementation of prehabilitation programs more challenging. Hospitals should be equipped with dedicated personnel, or ideally, physiotherapy facilities should be available near hospitals. Since this paper is essentially a meta-analysis with significant biases related to the heterogeneity of the studies considered, the inclusion and exclusion criteria of the articles selected from international literature are clearly explained. The study selection process was conducted using well-described and fully reproducible criteria for any group interested in replicating the work. The results are well described and highlight two key pillars: what we could call preoperative physiotherapy, and the involvement of the nutritionist (see doi.org/10.3390/nu17010188, which should be read and cited in the bibliography), since all oncology patients should be considered "de facto" malnourished or, more accurately, undernourished. In fact, we strongly recommend stating that, during the multidisciplinary team discussion after diagnostic workup and before determining the appropriate therapeutic pathway, the patient should also be assessed by a nutritionist to improve their condition ahead of any medical or surgical treatment. In the discussion section, we appreciated the inclusion of family members and caregivers, as well as the barriers encountered by researchers, which ultimately translate into the paper's biases. Psychological support is an excellent addition to prevent dropouts or non-adherence to programs. Congratulations on this first paper on the subject. As a final recommendation, consider shortening it slightly, as some parts—particularly in the results and discussion—are somewhat redundant. Excellent visuals, solid English, and an outstanding bibliography.
Response: We greatly appreciate the time and effort the reviewer dedicated to evaluating our manuscript, and we are thankful for the encouraging feedback.
In response:
- We have added a brief paragraph to the Introduction (L58–L62) discussing how screening programs have contributed to a shift toward an older colorectal cancer population. “This demographic shift is partly attributable to the widespread implementation of CRC screening programs. These programs allow for early detection and removal of tumors, which reduces the overall number of advanced cases. As a result, the remaining patient population tends to be older and often presents with more complex medical issues [22].
- We recognize the importance of nutritionist involvement in CRC-care and have added the recommended citation (DOI: 10.3390/nu17010188) to our Introduction (L49-51).
- To improve clarity, we have reviewed the Results and Discussion and removed minor redundancies.
